**RESEARCH**

# Turnover of strain-level diversity modulates functional traits in the honeybee gut microbiome between nurses and foragers

Gilles L. C. Baud[1], Aiswarya Prasad[1], Kirsten M. Ellegaard[1] and Philipp Engel[1*]

*Correspondence:
philipp.engel@unil.ch

[1] Department of Fundamental Microbiology, University of Lausanne, CH-1015 Lausanne, Switzerland

## Abstract

**Background:** Strain-level diversity is widespread among bacterial species and can expand the functional potential of natural microbial communities. However, to what extent communities undergo consistent shifts in strain composition in response to environmental/host changes is less well understood.

**Results:** Here, we used shotgun metagenomics to compare the gut microbiota of two behavioral states of the Western honeybee (*Apis mellifera*), namely nurse and forager bees. While their gut microbiota is composed of the same bacterial species, we detect consistent changes in strain-level composition between nurses and foragers. Single nucleotide variant profiles of predominant bacterial species cluster by behavioral state. Moreover, we identify strain-specific gene content related to nutrient utilization, vitamin biosynthesis, and cell–cell interactions specifically associated with the two behavioral states.

**Conclusions:** Our findings show that strain-level diversity in host-associated communities can undergo consistent changes in response to host behavioral changes modulating the functional potential of the community.

**Keywords:** Metagenomics, Gut microbiota, Strain diversity, Honey bee, Symbiosis, Social insects

## Background

Microbial communities are highly dynamic. They undergo compositional changes in response to environmental factors, which can profoundly influence their functions. For example, antibiotic treatment can disturb the human gut microbiota, altering its effect on the host metabolism and immune system, and facilitating the invasion of pathogens [1]. Such microbiota shifts are difficult to predict as they often occur irregularly and in different contexts. However, when linked to recurrent or predetermined environmental changes, microbial communities can undergo consistent compositional transitions, providing excellent opportunities to study ecological interactions

governing community shifts. For example, bacterioplankton communities in the ocean that depend on sunlight intensity [2–4], or gut microbial communities of animals that have a seasonal diet, or that hibernate during the winter months [5–8], turnover as a function of season. Similarly, the assembly of the human gut microbiota after birth follows a predictable pattern characterized by the succession of different microbial taxa during the first few months of life [9, 10].

Compositional changes associated with the turnover of microbial communities have mostly been studied at the genus- or species-level using 16S rRNA gene analysis. However, natural microbial communities usually harbor a large extent of strain-level diversity, with strains of the same species carrying distinct functions. In the human gut, strain-level differences can influence the metabolism of dietary compounds [11, 12] or drugs [13] or modulate their interactions with the host immune system [14]. Moreover, several recent studies have suggested that strains of the same species can exhibit different dynamics and govern the eco-evolutionary dynamics in microbial communities [15–18]. However, if natural microbial communities undergo consistent compositional transitions at the strain level in response to environmental changes is less well understood.

Honeybees (*Apis mellifera*) can be useful models for studying strain-level dynamics in host-associated microbial communities. They are eusocial animals that live in large colonies composed of a single queen, a few hundred male bees, and thousands of sterile female worker bees. They harbor a relatively simple, but highly specialized microbial community in their gut, composed of < 10 bacterial phylotypes (i.e., bacteria sharing > 97% sequence identity in the 16S rRNA gene) [19, 20]. Most of these phylotypes have diverged into sequence discrete populations (sequence clusters of > 95% ANI; hereafter referred to as species), each harboring a high degree of strain-level diversity and variation in functional gene content [21–26].

Using shotgun metagenomics, we have previously shown that while phylotypes, and species within phylotypes, co-exist in individual bees, strains within species segregate, suggesting that each worker bee harbors a distinct community at the strain level [22]. These differences in strain-level composition could not be attributed to any variation in age, time of sampling, or colony affiliation. However, in our previous study, only a few colonies were sampled, and bees were exclusively sampled from within the hive between September and February (i.e., autumn and winter) raising the possibility that most microbiota samples originated from nurse bees or long-lived winter bees. During the summer months, worker bees undergo an age-related behavioral maturation [27–29]. Young nurse bees feed on pollen and fulfill tasks inside the hive related to brood care, nest building, and food processing. However, when 2–3 weeks old, nurses transition to forager bees, which feed on honey, and collect pollen and nectar outside the hive. Besides the dramatic change in dietary intake, this behavioral maturation is accompanied by a series of physiological changes, such as endocrine secretion, fat body metabolism, and brain activity [30, 31]. Community profiling based on 16S rRNA gene sequencing has shown that the nurse-to-forager transition is associated with a consistent change in gut microbiota composition [32, 33]. While the microbiota of both bee types is dominated by the same bacterial phylotypes, the

composition of the community shifts with a marked decrease in the absolute abundance of most phylotypes in forager versus nurse bees.

Here, we tested if the gut microbiota of honeybees during the transition from nurse to forager bees also undergoes consistent compositional changes at the strain level. To this end, we analyzed shotgun metagenomics data from nurse and forager bees sampled across 15 colonies in Western Switzerland. Our results show that the two behavioral states harbor distinctive bacterial communities across colonies, characterized by consistent shifts in strain composition and functional gene content of major bacterial species. These findings show that strains are ecologically distinct units that undergo a predictable ecological transition when bees turn from nurses to foragers. We speculate that this microbiota shift is driven by nutrient availability disparities or physiological changes in the gut of foragers relative to nurses.

## Results

### Bacterial loads and species-level community composition differ between nurse and forager bees

To investigate compositional changes with genomic resolution in the honeybee gut microbiota in response to behavioral state, we collected metagenomic samples from larvae-feeding nurses and pollen foragers (Additional file 2: Table S1). Samples consisted of pooled honeybee guts to ensure representative sampling of the strain-level diversity for each colony. In total, we sampled 15 colonies from five different sampling locations (three each), to account for possible variation between sites and colonies, not related to behavioral state.

Due to the different dietary habits of the two behavioral states, the hindguts of nurses were usually pollen-filled, as indicated by the enlarged gut size, and the opacity and yellow color of the gut content. In contrast, the hindguts of foragers were smaller and contained mostly nectar based on their translucent appearance. Accordingly, nurse guts were on average $2.9 \times$ heavier than their forager counterparts ($p = 3.3e-09$, Wilcoxon signed-rank test, Additional file 1: Fig. S1).

To estimate the absolute abundance of bacteria relative to the amount of total DNA in each pooled gut sample, we performed qPCR on genomic DNA targeting the bacterial 16S rRNA gene. This revealed that the bacterial loads in the forager samples were on average $2.6 \times$ lower than the ones in the nurse samples ($p = 1.83e-04$, Wilcoxon signed-rank test, Fig. 1A), confirming previous results by Kešnerová et al. [33].

Shotgun metagenomics sequencing resulted in 58.4–76.7 million reads per sample. To compare the bacterial community composition of nurses and foragers, the metagenomic reads were mapped to a reference database of isolate genomes of strains from the gut microbiota of social bees, as in Ellegaard et al. [34] and Ellegaard and Engel [22]. The non-redundant database was composed of 198 isolate genomes and encompassed all major community members described to be present in the honeybee gut microbiota. Between 19.8 and 57.4% of all metagenomic reads mapped to the bacterial reference database. Of the remaining reads, 30.6–76.0% mapped to the host genome of *A. mellifera,* and only 4.2–20.8% of the reads could not be assigned (Additional file 1: Fig. S2A). Forager samples overall had a lower proportion of reads mapping to the bacterial database than the nurse samples (forager average 35.6%, nurse average 50.8%,

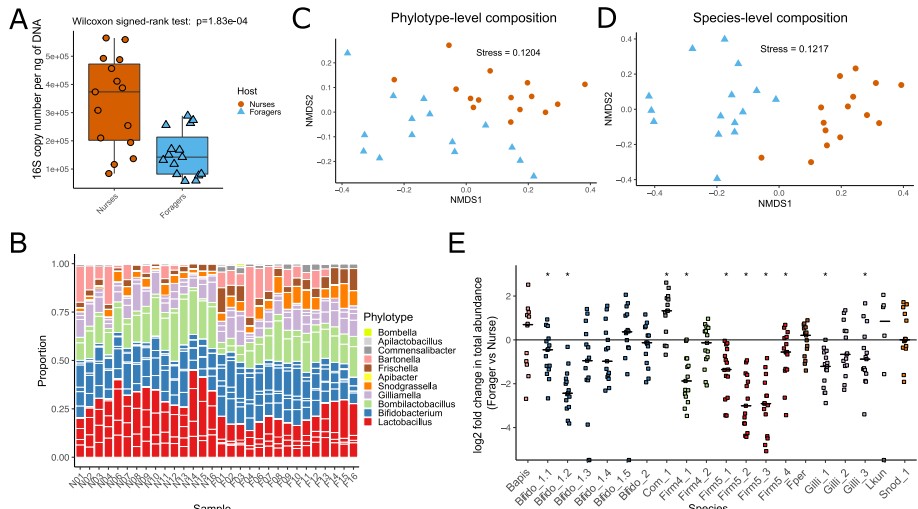

**Fig. 1** Community composition analysis of the gut microbiota of nurse and forager bees. **A** Boxplot of the bacterial loads of each gut represented by the number of bacterial 16S rRNA gene copies relative to 1 ng of total DNA. **B** Stacked bar plot of the bacterial phylotype composition for each sample. White separators within a phylotype delineate the different species. **C** NMDS plot of the samples' bacterial phylotype composition. **D** NMDS plot of the samples' bacterial species composition. **E** Log2 fold change in total normalized absolute abundance of bacterial species in forager samples relative to nurse samples from the same hive. Asterisks mark species with significant differences between nurses and foragers (Wilcoxon signed-rank test of the normalized absolute abundance of each species in the foragers against the nurses, *q*-value < 0.05). Data points for species of the same phylotype are shown in the same color. Horizontal bars represent medians

$p = 6.1e - 05$, Wilcoxon signed-rank test), which is in line with the difference in bacterial loads between the two behavioral states. Taxonomic profiling with mOTU2 [35] revealed that a large fraction of the unmapped reads originated from bacteria, especially from *Bartonella*, *Snodgrassella*, and *Lactobacillus* suggesting that some of the diversity of these core microbiota members is not covered in our reference database. However, also some accessory community members were detected among the unmapped reads such as *Hafnia*, *Klebsiella*, or *Enterobacter*, especially in nurse samples (Additional file 1: Fig. S2B).

We next determined the relative and absolute abundance of the major phylotypes and species in each sample based on the read coverage of single-copy core genes and the total number of 16 rRNA gene copies determined by qPCR. We will refer to different species within phylotypes using the previously established SDP (sequence-discrete population) numbering [22, 34], for reasons of readability but also because only for a subset of them proper species names have been established to date. A list with all SDPs and corresponding species names can be found in Additional file 2: Table S2. Both the phylotype and species-level composition were comparable to the community profiles detected in previous shotgun metagenomic analyses based on individual bees [22, 34]. Firmicutes (Bombilactobacilli and Lactobacilli) and Bifidobacteria dominated the community, and for the phylotypes *Bombilactobacillus*, *Lactobacillus*, *Bifidobacterium*, and *Gilliamella*, different species were detected in all samples (Fig. 1B, Additional file 1: Fig. S3). Consistent with previous findings that social bees harbor host-specific communities [34, 36], most reads mapped to the genomes of bacterial isolates from *A. mellifera*. Only a few

reads (0.15–3.1%) in nine samples mapped to bacteria from other bee species (including Lactobacilli isolated from *Apis cerana,* and Gilliamella isolated from bumble bees and *Apis cerana*; see Additional file 2: Table S3), which may be due to spurious mapping.

Non-metric multidimensional scaling (NMDS) using Bray–Curtis dissimilarity of the bacterial relative abundances separated nurse from forager samples at both the phylotype- (PERMANOVA, Host—$R^2 = 0.40279$, *p*-value = 0.001) and the species-level (PERMANOVA, Host—$R^2 = 0.45201$, *p*-value = 0.001, Fig. 1C and D). Samples are also clustered by location (Additional file 1: Fig. S4). However, it is important to note that our study was not designed to test for this, as only three colonies were sampled per location with two samples per colony.

In terms of absolute abundances (as determined by multiplying the proportions of each community member by the total bacterial biomass in each sample, see the "Material and methods" section), most community members were less abundant in foragers than in nurse bees (Additional file 1: Fig. S5), a direct consequence of the lower total bacterial biomass present in the gut of foragers (Fig. 1A). However, there were clear differences in the strength of these changes between community members, even among species of the same phylotype (Fig. 1E). For example, the total abundances of Lactobacilli Firm5_1 and Firm5_4 changed to a lesser extent (average log2 fold change = -1.45 and − 0.71) than those of Firm5_2 and Firm5_3 (average log2 fold change = − 2.82 and − 2.99) in foragers relative to nurses. This explains the consistent shift towards a less even species composition of this phylotype in foragers compared to nurses as determined with the Shannon diversity index (Wilcoxon signed-rank test, *p*-value = 8.8e − 06, Additional file 1: Fig. S3). Likewise, Bifido_1.2 was more affected (average log2 fold change = − 2.32) than any of the other species of the *Bifidobacterium* phylotype (average log2 fold change across species of the *Bifidobacterium* phylotype = − 0.37) by the behavioral state transition of the host (Additional file 1: Fig. S3). Notably, four community members (Bapis, Snod_1, Com_1 and Lkun) did not change in absolute bacterial load between nurses and forager samples, suggesting that they are not affected in their abundance by the behavioral transition.

In summary, although the gut microbiota of forager and nurse bees were composed of the same community members, we found consistent differences in the absolute abundance of many of them, not only at the phylotype level as previously shown [32, 33], but also at the species level. Moreover, some species of a given phylotype were more strongly affected than others suggesting species-specific responses to the nurse-to-forager transition.

### Consistent differences in strain-level composition between nurse and forager bees

Previous studies have shown that most honeybee gut microbiota species harbor high levels of strain diversity and that the strain composition differs between individual bees [22, 34]. To assess strain-level differences between nurses or foragers, we mapped the metagenomic reads against a reduced reference database containing one genome per species and quantified the number of polymorphic sites in the reads mapped onto the single-copy core genes of these reference genomes.

We detected between 1.85 and 11.19% polymorphic sites per species per sample, with Firm4_2 showing the lowest, and Firm5_4 the highest average proportion of

polymorphic sites per sample (Fig. 2A). These differences in proportions of polymorphic sites among community members are comparable to those previously found by Elle-gaard et al. [34] and Ellegaard and Engel [22]. To assess the total amount of strain-level diversity detected per species across all samples, we calculated cumulative proportions of polymorphic sites as a function of the number of bee samples. Firm4_2 showed the lowest (4.91%) and Bifido1_1 the highest (12.8%) level of total diversity across all samples (Fig. 2B). In all cases, we observed that the slope of the curves rapidly decreased with the addition of new samples, indicating that many of the polymorphic sites are shared across samples.

When comparing the two behavioral states, we found that five of the 17 analyzed species showed higher strain-level diversity in nurses than in foragers (Wilcoxon signed-rank test, Firm4_1: $q$-value $= 4.20e-3$, Firm5_1: $q$-value $= 6.45e-3$, Firm5_2: $q$-value $= 1.08e-2$, Firm5_3: $q$-value $= 3.56e-4$, Gilli_1: $q$-value $= 6.45e-3$) whereas only two species showed lower strain-level diversity in nurses than in foragers (Wilcoxon signed-rank test, Gilli_2: $q$-value $= 2.91e-2$, Snod_1: $q$-value $= 2.91e-2$, Fig. 2A). The

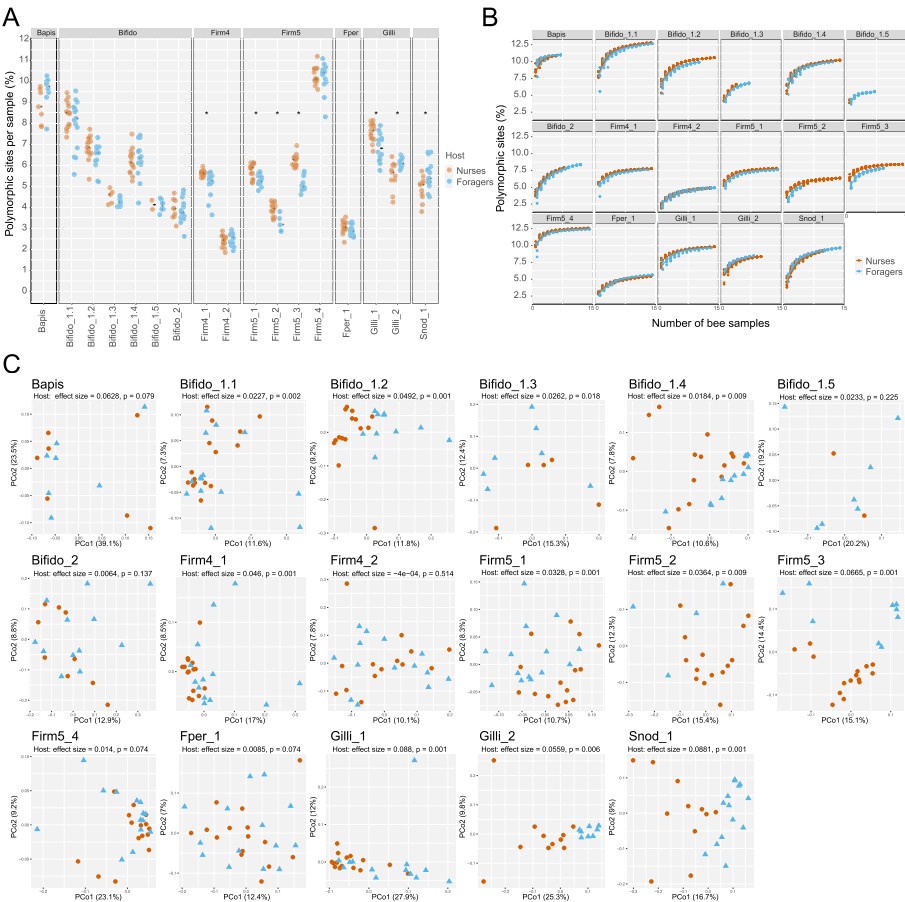

**Fig. 2** Differences in strain-level diversity between nurse and forager bees. **A** Proportion of polymorphic sites in the single-copy core genes for each bacterial species, expressed as a percentage of the total single-copy core genome length. Asterisks symbolize a *q*-value < 0.05 for the Wilcoxon signed-rank test between the proportion of polymorphic sites between nurse and forager samples. **B** Curves of the cumulative proportions of polymorphic sites for each species. **C** Principal coordinate analysis plots based on the Jaccard distance shared polymorphic sites between each pair of samples for each of the species

forager samples had on average 2.6 times lower bacterial load than nurse samples, but on average only 1.5 times smaller number of reads per sample. This means that a larger fraction of the total bacterial population was analyzed for forager relative to nurse samples, a difference that could potentially affect diversity estimates. However, no significant correlation was found between the proportion of polymorphic sites detected and the read coverage of our samples (Additional file 1: Fig. S6), suggesting that we had sufficiently deep coverage across the study.

Next, we assessed the difference in strain profiles between forager and nurse samples by comparing the distribution of single nucleotide variants (i.e., alleles) at the detected polymorphic sites for each species across samples. To do so, we computed the Jaccard distances between all sample pairs based on the number of shared polymorphic sites. Principal coordinate analysis of these similarity indices resulted in clustering of the samples by host behavioral state for 11 of the 17 analyzed species (PERMANOVA, $p$-value < 0.05, Fig. 2C), including all species of Bifido_1 and Firm5, except for Bifido_1.5 and Firm5_4, Firm4_1, Gilli_1, Gilli_2, and Snod_1. Notably, we also observed clustering by location in the principal coordinate analysis (PERMANOVA, $p$-value < 0.05, Additional file 1: Fig. S7). However, as already stated before, our study was not designed to test for the effect of location as two samples per location came from the same colony.

In summary, our analysis shows that nurse bees and foragers have distinct strain profiles for many bacterial species both in terms of the amount of diversity and type of variants detected.

### Intraspecific differences in functional gene content between nurse and forager bees

Previous studies have shown that most species of the honeybee gut microbiota harbor large accessory gene pools and that individual bees vary in terms of the functional gene content encoded in a given species [22]. Because nurses and foragers have different physiology and diet, we thus hypothesized that the observed changes in strain-level composition between the two behavioral states may be driven by functional differences among strains.

To identify strain-specific genes that may be enriched in one behavioral state or the other, we compared the read coverage of all genes assigned to a given species. Strain-specific genes should have a lower read coverage than the single-copy core genes of a given species, and those that predominate in one behavioral state relative to the other should show a consistent coverage discrepancy between forager and nurse samples. To identify such genes, we assembled all reads not mapping to the host genome and binned the resulting contigs into bacterial species bins using our reference database (see the "Material and methods" section for details and Additional file 1: Fig. S8). Then, we clustered the ORFs encoded in the binned contigs into species-specific ortholog groups (OGs) and identified those OGs that showed consistent coverage discrepancy between forager and nurse samples. In total, 81.7% (65.3–90.9% per sample) of all assembled bases could be binned and 83.4% (73.2–86.1% per sample) of the non-host reads mapped to the binned contigs indicating that we had included most of the bacterial diversity present in the samples. Out of 45,863 species-specific OGs, 1551 revealed significant differences between the nurse and forager samples ($q$-value < 0.05, Wilcoxon signed-rank test, and absolute log2 fold change > 1, Additional file 2: Table S4), which indicated that these

genes were more abundant in one host behavioral state versus the other. Notably, the number of differential abundant OGs substantially varied between species ranging from 15 to 231 (Fig. 3A, Additional file 2: Table S4). For example, in Bifido_1.1, we detected very few consistent gene content differences between foragers and nurses (6 and 9 OGs, respectively). This species had the highest proportion of polymorphic sites across samples, but showed no clustering of the samples by behavioral state based on the number of shared polymorphic sites (Fig. 2C). The largest differences in gene content between nurses and foragers were found for three *Lactobacillus* Firm5 species (Firm5_2: 184 OGs, Firm5_3: 131 OGs, Firm5_4: 177 OGs), two Gilliamella species (Gilli_1: 64 OGs, Gilli_2: 161 OGs), Snod_1 (197 OGs), and Bifido_1.2 (159 OGs). Consistent with these findings, all of these species, except for Firm5_4, also showed clustering by behavioral state based on the number of shared polymorphic sites (Fig. 2C). Interestingly, a larger fraction of the state-specific OGs were found in foragers than in nurses (in total 1019 versus 445 significantly different OGs, respectively).

To determine the functions encoded by the behavioral state-specific gene content, we annotated all OGs using eggnog and used COG categories as a first level of functional characterization. Of all behavioral state-specific OGs, 33.1% had no COG annotation, while 17.3% of them fell into the COG category S (unknown function). For the rest of the significant OGs, we categorized 20.6% of them as metabolism-related; 11.7% as linked to cellular processes, signaling, and structure; and 18.3% as information storage and processing (Fig. 3B). Among the significant categorized OGs (excluding COG S), category G (carbohydrate metabolism and transport) was the most important category, representing 11.4% of all significant OGs. However, the proportion of different COG categories contributing to the significant OGs varied between species (Fig. 3B). In case of Firm4_1, COG category H (coenzyme metabolism) was specifically enriched in the behavioral state-specific OGs relative to the total number of OGs per species (Fisher's exact test $q$-value $= 2.5e − 06$, Additional file 2: Table S5).

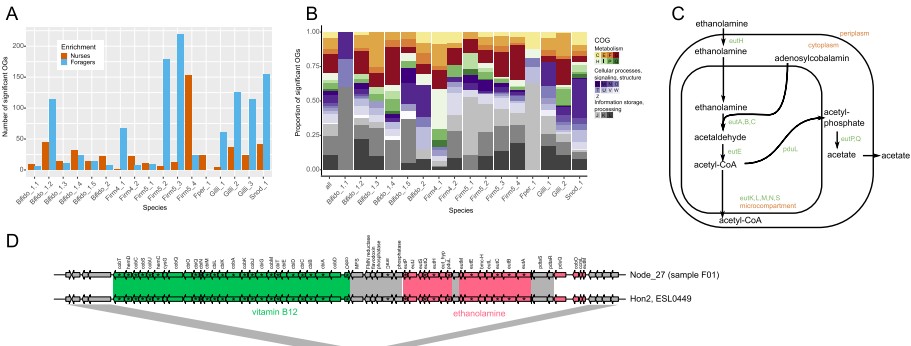

**Fig. 3** Differences in functional gene content within species of the gut microbiota between nurse and forager bees. **A** Number of orthologous groups (OGs) significantly differing in coverage for each of the species. **B** Repartition of the COG categories of the significant OGs, nurse-enriched and forager-enriched included. OGs that were not categorized or categorized as COG S were not included in the plot. **C** Figure adapted from [37]. Possible ethanolamine utilization pathway enabled by the operon shown in **D**. **D** Genomic island containing the ethanolamine utilization and vitamin B12 biosynthesis genes, and its presence in the isolate genomes of the bacterial genome database. Genes enriched in the forager samples are in bold and flagged with an asterisk

Behavioral state-specific OGs of COG category G included many phosphotransferase system genes, in particular in *Lactobacillus* and *Gilliamella* species, but also glycoside hydrolases, sugar-binding, and metabolism proteins. In most species, these OGs were more abundant in foragers relative to nurses, except for Bifido_1.2 and Firm5_3, where the enrichment was in the nurse samples, or spread equally between nurse and forager samples (Additional file 2: Table S4).

The overrepresentation of the COG category H in Firm4_1 corresponded to 27 OGs involved in the biosynthesis of cobalamin (vitamin B12) and adenosylcobalamin (coenzyme B12), which had a higher coverage in forager than in nurse samples (see Additional file 2: Table S4). In the same species, we also found 18 genes for the utilization of ethanolamine among the OGs enriched in foragers versus nurses. Processing of ethanolamine by the enzyme ammonia-lyase is dependent on adenosylcobalamin (Fig. 3C). To further explore the possible link between the identified cobalamin biosynthesis and ethanolamine utilization OGs, we analyzed the available isolate genomes of Firm4. Homologs of the cobalamin biosynthesis and ethanol utilization pathways were only present in two (Hon2 and ESL0449) of the six Firm4_1 strains (Fig. 3D) confirming that these genes are strain-specific. Moreover, the genes of both pathways were encoded in the same genomic island, integrated into a conserved chromosomal region, providing further evidence that they are functionally linked and have likely evolved together.

Another interesting set of behavioral state-specific OGs was identified in Gilli_1, Gilli_2, and Snod_1, and encoded various components of Type VI secretion systems (T6SSs) involved in interbacterial killing. This included genes annotated as effector proteins with PAAR or RHS domains and immunity proteins. In Gilli_2, several genes encoding the structural components of an entire T6SS were among the behavioral state-specific OGs. They clustered in a genomic island which was present in only one (strain A7) out of 17 strains of Gilli_2 in our reference database (Additional file 1: Fig. S9). Strikingly, all these behavioral state-specific T6SS-related OGs were more abundant in forager than in nurse samples.

In Snod_1, we also found other surface structure-related functions than T6SS among the behavioral state-specific OGs, including genes encoding major subunits of Type IV pili, RTX toxins, and proteins involved in capsular polysaccharide biosynthesis (Additional file 2: Table S4).

## Discussion

There is accumulating evidence that natural microbial communities harbor a high extent of strain-level diversity [38–41]. How such fine-scale diversity varies across space and time in nature is a key question in microbial ecology and evolution [15, 42, 43].

Here, we find that the strain composition of the honeybee gut microbiota undergoes a consistent ecological turnover when bees transition from nurses to foragers. The single nucleotide variant profiles of several community members clustered by behavioral state. Moreover, for most community members, we found consistent differences in intraspecific gene content between nurses and foragers, which included functions related to nutrient utilization, vitamin biosynthesis, and cell–cell interactions.

In the human gut microbiota, strain-level variation in individual community members has been linked to diet [44] or health state [43, 45–47]. Our results show that a

radical shift in host lifestyle can change the strain composition of a large fraction of the gut microbiota, demonstrating that strains represent ecologically distinct units and emphasizing the importance of studying diversity in host-microbe associations at the strain-level.

Previous shotgun metagenomics studies have linked variation in species- and strain-level diversity in the gut microbiota to host traits, such as age, colony origin, or genotype, but not to behavioral state [22, 34, 48–50]. Some of these studies analyzed individual bees while others focused on pooled gut samples. In the current work, we analyzed pooled samples of 20 bees rather than individual bees. The advantage of pooling is that a larger number of colonies can be surveyed, increasing the statistical power of the study. However, pooling precludes the detection of variants present at low prevalence across the sampled bees explaining the overall lower levels of polymorphic sites observed in our samples as compared to previous studies [22, 34]. However, rare strains have likely a minor impact on the whole community or the host.

Another possible limitation of our study is that the sampled nurse and forager bees were not age-matched. As foragers are usually older than nurses, age could have been a confounding factor in our study. However, our previous findings have shown that the age of bees has little impact on the strain level composition of the gut microbiota [22]. Therefore, we are confident that most of the observed differences are due to the nurse to forager transition and the associated changes in dietary intake, physicochemical gut conditions, or host physiology.

In contrast to nurses, foragers do not consume pollen. This is likely the reason why foragers were found to harbor lower bacterial biomass in the gut than nurses and exhibit a consistent decrease of *Bombilactobacillus*, *Lactobacillus*, and *Bifidobacterium* relative to the other phylotypes. This was already reported by two previous studies based on 16S rRNA gene analysis [32, 33], one of which used age-matched forager and nurse bees, corroborating the results of our shotgun metagenomics analyses. Pollen constitutes an important nutrient source for bee gut-associated *Bombilactobacillus*, *Lactobacillus*, and *Bifidobacterium* explaining their decrease in absolute abundance. All three phylotypes harbor genes to degrade pollen-derived carbohydrates [21, 26, 51, 52], and they readily grow in minimal media supplement with pollen extract or entire pollen grains [53, 54]. Moreover, experimentally colonized bees kept on a sugar water diet harbor substantially fewer bacteria of these phylotypes in their gut than bees that also receive pollen [33, 53].

As previous studies were based on 16S rRNA gene sequencing, their analyses were limited to the phylotype level. Hence, a key finding of our study is that not only phylotypes, but also closely related species within phylotypes, and even different strains of the same species respond differently to the nurse-to-forager transition. For example, one species of *Lactobacillus* Firm5 was similar in total abundance between foragers and nurses, while others significantly differed (>8 times). Likewise, some species showed consistent changes in strain composition between nurses and foragers, while others did not. This suggests that some species and strains are more successful than others in thriving in the gut environment of one state versus the other. The reasons underlying these differences can be manifold. Some species and strains may be adapted to utilize specific pollen compounds and hence have an advantage in the nurse gut. Others may not depend on pollen nutrients for growth that much or may be better at withstanding physicochemical

changes in the gut as bees transition from one state to the other, explaining why they persist in the forager gut. Also, the fact that we find changes in strain-level diversity in opposite directions between the two host states suggests that different ecological processes are at play in different species. For example, in one species, the reduced nutrient availability in the forager gut may hinder strains to overgrow and dominate the community, resulting in decreased total abundance but increased strain-level diversity in foragers. For another species, the nutrient-rich pollen diet may expand ecological niches or open new ones that can be colonized by functionally divergent strains explaining the observed increase in strain-level diversity in nurses relative to foragers.

If the observed shifts in strain composition were due to changes in the ecological niches in the gut, we would expect to find functional differences between strains of the same species dominating in nurses versus foragers. Previous studies have successfully identified such condition-specific genes within host-associated bacterial species by comparing metagenomic datasets [17, 38, 44]. For example, strains of the human gut symbiont *Prevotella copri* harbored distinct gene repertoires for drug metabolism and complex carbohydrate degradation in Western and non-Western individuals [44]. Likewise, strains of an intracellular symbiont of *Bathymodiolus* deep-sea mussels showed substantial variation in gene content for nutrient and energy acquisition indicating adaptation to the local conditions present in the environment they were sampled from [38].

We detected strain-specific gene sets (i.e., OGs) associated with one of the two behavioral states in almost all species of the gut microbiota. Carbohydrate transport and metabolism functions were prevalent among the behavioral state-specific gene content, which was somehow expected as most species carry large accessory gene sets of this functional category in their genomes [22, 51, 52]. Moreover, previous studies have shown that carbohydrate transport and metabolism are important for niche partitioning of diet-derived carbohydrates in *Lactobacillus* and *Bifidobacterium* and that many of these genes are differentially expressed when bees are fed pollen as compared to sugar water as diet [26, 53].

Somewhat more surprising was the finding that vitamin B12 biosynthesis and ethanolamine utilization genes of *Bombilactobacillus* were specifically enriched in foragers versus nurses. Some of these genes have already been found to be variably present among individual bees in our previous study [22], but they could not be linked to any host differences. Vitamin B12 is an important co-factor that cannot be synthesized by honeybees (or other animals) [55], but needs to be taken up from the diet or synthesized by bacterial symbionts [56]. It is possible that *Bombilactobacillus* strains synthesizing this vitamin compensate for the lack of vitamin B12 in the pollen-deprived diet of foragers. However, a more plausible explanation for the increased abundance of these genes in foragers may be that vitamin B12 is an essential co-factor for a major enzyme of the ethanolamine utilization pathway [55]. Ethanolamine is derived from the membrane phospholipid phosphatidylethanolamine and constitutes an important bacterial nitrogen and carbon source in the animal gut due to the high turnover of the host epithelium and the microbiota [57]. It is possible that ethanolamine accumulates in the forager gut due to the presence of dead bacterial material or increased epithelial shedding, providing a behavioral state-specific nutrient niche for the gut microbiota. The fact that the genes for utilizing ethanolamine are strain-specific indicates that they are costly to maintain,

and that host state-specific nutritional niches may contribute to the maintenance of strain-level diversity in the gut microbiota.

We also found genes mediating bacteria-host interactions and interbacterial killing among the behavioral state-specific gene content. Of particular interest was the finding of several T6SS genes associated with foragers in two of the three Gilliamella species and in Snodgrassella. The primary niche of these bacteria is the ileum, where they form multispecies biofilms on the host epithelium [58]. Diverse T6SS gene repertoires have been identified in almost all sequenced strains of Gilliamella and Snodgrassella indicating that they play key roles in interbacterial competition and niche establishment for these bacteria [59, 60]. Our findings suggest that the forager gut environment selects strains with expanded capabilities in interbacterial warfare, possibly as a consequence of the increased competition for nutrients and space. In Snodgrassella, we also found genes encoding other surface structures (Type IV pili, LPS modifying enzymes, RTX toxins) to be enriched in either nurses or foragers. Some of these genes are known colonization factors of Snodgrassella [61], likely because they enable host attachment and biofilm formation. However, why these functions are specifically enriched in one of the two behavioral states remains to be determined.

## Conclusions

In summary, using shotgun metagenomics analysis, we show that the gut microbiota of honeybees undergoes consistent shifts in strain-level diversity when bees transition between two major behavioral states (i.e., from nurses to foragers). These strain-level differences are associated with changes in intraspecific gene content of several bacterial species suggesting adaptation to host state-specific niches. The possibility of carrying out gnotobiotic bee experiments provides excellent opportunities for future research to probe the functional relevance of the link between strain-specific gene content and host state. Since the association of social bees with their microbiota is evolutionarily ancient, it will also be interesting to expand these analyses to closely related bee species which show similar division of labor and whose microbiota has diversified to similar extents as in *A. mellifera*.

## Material and methods

### Sampling

Adult female worker bees of *A. mellifera* were collected over the Summer 2019 from 15 different colonies in 5 different locations across Western Switzerland (Additional file 2: Table S1). For each sampled hive, we collected 20 larvae-feeding nurses and 20 pollen foragers, in order to encompass most of the strain-level diversity within each colony. Samplings were performed between 7 and 9 AM, when worker bees were beginning to return from their foraging trips. Larvae-feeding nurses were identified as bees putting their heads in larvae-containing cells. Pollen foragers were identified as bees returning to their hive with pollen pellets on their corbiculae. The bees were anesthetized using $CO_2$ and put on ice. They were then dissected in sterile $1 \times$ PBS to extract the entire hindgut (pylorus, ileum, rectum) with tools sterilized in 70% ethanol. Gut content was visually inspected to be consistent with the life stage, i.e., nurse guts had to be pollen-filled and forager guts had to be nectar-filled. Bees for which the gut content could not be verified

were excluded. Finally, the guts were pooled by hive of origin and host phenotype (nurse and forager, respectively) (Additional file 2: Table S1).

### DNA extraction

DNA extraction from the pooled gut samples was performed using the Qiagen All-prep Powerfecal DNA/RNA extraction kit (Qiagen) following the kit handbook except for the bead-beating steps. After dissection, the pooled guts were transferred into the kit's bead-beating tubes. Lysis buffer and DDT were added following protocol instructions, and samples underwent two bead-beating steps of 4 s at 6 m/s on a FastPrep-24 5G bead-beater (MP Biomedicals). Gut samples with a total weight over 220 mg were diluted to reach a concentration of 220 mg in 650 µl lysis buffer + 25 µl DDT; then, the excess volume was removed (i.e., for a gut weight of 440 mg, 337.5 µl was removed from the sample, and an identical volume of lysis buffer + 3.8% DDT 1 M was added to it). Subsequently, all samples underwent a third bead-beating step for 45 s at 6 m/s. All the remaining steps were performed following the manufacturer's protocol. DNA was diluted in $2 \times 30$ µl Buffer EB, its concentration and quality were assessed with Qubit and Nanodrop, and samples were stored at $-20$ °C until further processing.

### Quantitative real-time PCR

We assessed the bacterial loads of the DNA samples using quantitative real-time PCR (qPCR) following the methods of Ellegaard et al. [34]. We targeted both the V3-V4 region of the 16S rRNA bacterial gene and the actin gene of *A. mellifera* using the primers published by Kešnerová et al. [54]. We performed each qPCR reaction using a reaction of 1 µl extracted DNA (10 ng/ µl), 3.2 µl nuclease-free water, 5 µl SYBR Select Master Mix (Life Technologies), 0.2 µl forward primers (5 µM), and 0.2 µl reverse primers (5 µM) in 96-well plates (MicroAmp Fast 96-well Reaction Plate, Applied Biosystems for Life Technologies). The reactions were performed in triplicates on a QuantStudio 5 Real-Time PCR System (Thermo Fischer Scientific) using the parameters from Kešnerová et al. [54].

The data was extracted using the QuantStudio Design & Analysis Software v1.4 (Applied Biosystems) and exported in csv format.

We computed the absolute number of each gene copy using the formula:

$$\text{Gene copy number} = 10^{\frac{C_t - \text{Intercept}}{\text{Slope}}}$$

Intercept and slope parameters were previously determined by our group [54].

In order to compare the microbial gut loads of the different samples, we normalized the number of 16S rRNA gene copies by the total DNA yield per sample. Normalization by the number of host actin gene copies gave similar results. However, foragers had more actin gene copies than nurses. Therefore, we did not use this type of normalization in the final analysis (see Additional file 1: Fig. S10). The figures were generated using R (version 4.0) and the ggplot2 library.

### Shotgun metagenomics sequencing

The libraries were prepared with the Illumina Nextera Flex library kit (Illumina) with unique dual indices (UDI) and sequenced on a HiSeq 4000 instrument (PE150) at the

Genomic Technologies Facility (GTF) of the University of Lausanne. All samples were multiplexed together and sequenced on three lanes. The quality of the reads was assessed with FastQC (v0.11.4, Babraham Institute) and subsequently trimmed and filtered to remove low-quality sequences and short reads using the tool Trimmomatic v0.35 [62] with settings PE ILLUMINACLIP:NexteraPE-PE.fa:2:30:10 TRAILING:20 SLIDING-WINDOW:4:20 MINLEN:50.

### Reference genome database

The reference genome database containing bacterial isolate genomes was constructed in the same way as in Ellegaard and Engel [22] and Ellegaard et al. [34]. A total of 198 high-quality genomes of strains isolated from the gut of honeybees and other social bee species such as bumble bees and other *Apis* species were included in the database. Strains having more than 98.5% pairwise ANI with other already included strains were not added to the database to limit redundancy. The pairwise genomic ANI was computed using fastANI [63]. In general, multiple genomes were included for each bacterial phylotype as listed in Additional file 2: Table S2. Each genome includes a fasta file of nucleotide sequences of all contigs (.fna), nucleotide sequences (.ffn), and amino acid sequences (.faa) of predicted genes and concatenated (.fna) files where the contigs were concatenated into one sequence for each genome. Bed files were made to summarize the updated positions of genes in the concatenated files which were used for further analysis. We predicted gene families of homologous genes using OrthoFinder (v2.3.5, [64]) and identified all single-copy core genes, i.e., genes present in one copy in all genomes of a given phylotype. The resulting files listing each single-copy core orthologous group and the constituent genes of each genome of that phylotype are included in the reference genome database along with reduced bed files for each genome indicating the location of these core genes to be used for further analysis. The total length of these single-copy core genes relative to the total genome length of a given species is given in Additional file 1: Fig. S11.

### Community composition analysis

The community composition analysis was based on the work of Ellegaard and Engel [22]. We mapped the reads using bwa mem (v0.7.15-r1142-dirty) [65] against the reference genome database and used the samtools (v1.9) view −F $0 \times 800$ command [66] to remove chimeric alignments. Reads mapping to multiple locations were not included as bwa chooses one of the many alignments as primary and only includes those in the output. Alignments with 50 matches or fewer were removed using a custom perl script that parses alignment length from the CIGAR string. The read coverage information was extracted for each gene using the command samtools bedcov and summed up across all orthologs of each single-copy core gene family for genomes coming from the same species. Since genes located close to the origin of replication had higher coverage than genes located close to the terminus (as an indication of replicating bacteria [67]), the coverage at the terminus was inferred. To this end, the coverage of each gene family was plotted relative to its genomic position using one selected reference genome per species and a segmented regression line was fitted. The relative abundance of each species in each sample was determined by dividing the terminus coverage by the summed

terminus coverage of all species, respectively. For the species Bifido_1.1, Bifido_1.2, and Bifido_1.4, the segmented lines did not show a proper fit (terminus location inferred far from estimated breakpoint, or coverage at *Ori* lower than *Ter*). In these cases, the median coverage of all single-copy core gene families was used as an estimate for the relative abundance of that species.

### Strain-level analysis

To quantify strain-level diversity within species, we followed the approach established in Ellegaard and Engel [22]. In short, we mapped the metagenomic reads using bwa mem to a reduced version of the reference genome database, including only one representative member per species (see Additional file 2: Table S2). Then, we used samtools -F $0 \times 800$ to remove chimeric alignments. Reads mapped with less than 50 matches registered in their CIGAR were also removed, as well as those whose number of mismatches was higher than 5 to reduce errors in variant calling. Only species with more than $20 \times$ terminus coverage were included. Genes with less than $10 \times$ coverage as estimated in the community profiling pipeline were excluded. To perform variant calling, we used the tool FreeBayes (v1.3.1–19-g54bf409) [68] with options -–haplotype-length 0, --min-alternate-fraction 0.1, –-min-coverage 10, -C 5, -I, -X, -u and –-pooled-continuous. We used the tools vcfbreakmulti and vcfallelicprimitives from the vcflib library [69] to split complex observations into multiple lines. The bed files containing gene positions of single-copy core genes of each genome were subset to obtain the corresponding locations to be considered for each species. These filtered bed files were used to focus downstream analysis on the phylotype-core region for each species.

We computed the fraction of polymorphic sites for each bacterial species using the number of identified sites divided by the cumulative length of all single-copy core genes. An allele was considered detected if there was more than one read supporting the allele and the allele was at a frequency of at least 0.01. A position was considered polymorphic if more than one allele was detected within the metagenomic reads of that sample mapped to the reference regardless of what allele was in the reference genome. We also computed the cumulative number of polymorphic sites across samples for each species using 10 random orders of samples each time.

To assess the similarity between samples in terms of strain composition, we used the Jaccard index based on the proportion of shared polymorphic sites, i.e., sites that were polymorphic in at least one of the samples in each pair and where each sample had an allele and an alternate allele shared and at a frequency above the detection threshold (0.01) were considered shared between the two samples. So, for example, if the reference genome has the allele T at a given position, but sample 1 has the alleles C and G and sample 2 has the alleles T and C, then the two samples would share one allele (C) and they would have two non-shared alleles. This site would not be counted as shared. On the other hand, if sample 1 had the alleles C and T the polymorphic site would be considered shared with sample 2. This was implemented in a custom python script as cases where the site was polymorphic in at least one sample in the pair were scored and shared positions were those with an intersection > 1 in the set of detected alleles in the sample pair. The distance between all sample pairs was computed as $Distance(S1, S2) = 1 - Shared\ fraction(S1, S2)$ where shared fraction for a given pair of

samples was calculated as the number of shared polymorphic sites divided by the number of sites polymorphic in at least one of the two samples. The matrix of pairwise distances between the samples was plotted using principal coordinate analysis to identify samples that share similar polymorphic sites. We computed the principal coordinates using the ape R library (v5.6–2).

### Functional gene content composition

For the functional profiling of the gene content of the microbiota, we first extracted all the reads which mapped to the bacterial database, plus those that did not map to the host genome (*A. mellifera* genome (PRJNA471592, version Amel_Hav3.1), using the picard tools (v2.7.1-SNAPSHOT) SamToFastq with parameters VALIDATION_STRINGENCY = SILENT [70]. We merged those reads for each sample and assembled them using SPAdes [71] with parameters --meta and -t 20. The obtained contigs were filtered to keep only the ones with a length of at least 500 bp and a k-mer coverage of at least 1.

We predicted the open-reading frame (ORF) positions using Prodigal (version 2.6.3, Hyatt et al.) [72] with parameters -m -p meta. The ORFs with the flag partial = 00 or length smaller than 300 bp were filtered out. We annotated the remaining ORFs using eggNOG (v1.0.3–40-g41a8498) [73, 74] with the script emapper.py and the options --annotate_hits_table --no_file_comments --cpu 20.

To assign the contigs to the major bacterial species of the bee gut microbiota, we blasted the predicted ORFs against a nucleotide database of all ORFs from our bacterial database using the blastn tool (version 2.2.31 +) [75] (*e*value < 1e − 05, pident > 50 and qcovs > 50). Contigs were assigned to a given species when at least 80% of the ORFs were mapped against genes of that species, as well as when at least 10% of the ORFs were mapped against genes of that species and no mapping was found for any other species. We filtered out all non-unanimous contigs (15,198 contigs out of a total of 257,015 (5.9%), representing 13.6% of the bp).

For each species, we used the amino acid sequences from the ORFs and the genes from the database strains to compute orthologous groups (OG) using OrthoFinder [64]. We created a database of all ORF's nucleotide sequences and mapped the reads against it using bwa mem (version 0.7.15-r1142-dirty) [65]. We removed chimeric alignments using samtools view option −F 0 × 800 (version 1.9) [66]. Alignments with less than 50 matches were also removed. For each sample, we extracted the coverage using the samtools coverage command (v1.15) and summed the coverage for the ORFs belonging to the same OG. We normalized the coverage information for every sample and every species using the OGs containing single-copy core genes: for each species of each sample, the summed coverage of all single-copy core OGs was set to the same number (10000). To determine which OGs were underrepresented or overrepresented in either the nurses or the foragers, we computed the paired Wilcoxon signed rank test for each OG based on the normalized coverage values and then used *q*-values to correct for multiple comparisons. We also computed the fold-change by dividing the mean forager normalized coverage value by the nurse counterpart. To be considered as significantly different between the treatments, an ORF had to have both an absolute log2 fold change of at least 1, and a *q*-value < 0.05. We finally annotated the ORFs belonging to the significant OGs using the

tool eggnog-mapper (1.0.3–40-g41a8498) [73, 74]. The whole analysis workflow is summarized in Additional file 1: Fig. S8.

## Statistics and data visualization

We performed the statistical analyses in R (v4.2.1) using the libraries ape (v5.6–2), data.table (v1.14.8), and vegan (v2.6–4) and in python (v3.10.11) using the package scipy (v1.11.1). We used the package qvalue (v2.28.0) to adjust *p*-values to q-values where stated. Shannon indices were calculated using the vegan package. For PERMANOVA, we used the function adonis2 from the vegan package using the model "Distance ~ Host + Location" with 999 permutations and set to estimate the marginal effects of the terms. $R^2$ values were adjusted using the function adonisOmegaSq implemented in the MicEco package (https://github.com/Russel88/MicEco, commit 91f8e6f).

We generated all plots using R and the following R libraries: ggplot2 (v3.3.6), reshape (v0.8.9), gridExtra (v2.3), ggrepel (v0.9.1), RColorBrewer (v1.1–3), and scales (v1.2.1) and in python (v3.10.11) using the package matplotlib (v3.7.1) and seaborn (v0.12.2). Figures have been generated using Inkscape (v1.1) or Adobe Illustrator. A comprehensive list of all packages used and their version is available in the GitHub repository (https://github.com/Aiswarya-prasad/Publication_Baud_metaG_NvsF_2023) [76].

## Supplementary Information

---

**Additional file 1: Fig. S1.** Per sample average gut mass, computed as each sample's gut pool mass, divided by the number of guts in the pool. **Fig. S2.** (A) Stacked bar plot of the proportions of reads mapping to the bacterial genome database and to the A. mellifera genome, as well as non-mapped reads, for each of the samples. (B) Taxonomic profiling of the unmapped reads at the species-level with mOTU2. Colors correspond to genera. Percentages below each distribution show fraction of unmapped reads in each sample (also see panel A). **Fig. S3.** Stacked bar plots of the proportions of the different species and corresponding Shannon diversity index across nurse and forager samples for the four multi-species phylotypes (A) Bifidobacterium, (B) Lactobacillus, (C) Bombilactobacillus, and (D) Gilliamella. **Fig S4.** NMDS plot of the samples' bacterial phylotype composition (A) and species composition (B). Samples are colored according to sampling location. PERMANOVA results: Phylotype-level, Location – $R^2$=0.40279, p-value=0.001; Species-level, Location – $R^2$=0. 40642, p-value=0.001. **Fig. S5.** Absolute abundance of the different species detected in the samples approximated by the normalized estimated number of 16S rRNA gene copies. Asterisks indicate q-values < 0.05. **Fig. S6.** Pearson correlation coefficient and non-corrected p-value for the correlation of the percentage of polymorphic sites per species with the coverage of the species normalized by the bacterial load for each sample. **Fig. S7.** Principal Coordinate Analyses of the Jaccard distance between all samples for the different species based on proportions of shared polymorphic sites. The color scheme shows information on the sampling sites of the different samples. The connecting lines indicates the two samples from the same colony. **Fig. S8** Workflow used for the functional gene content analysis. **Fig. S9.** Genomic island of Gilli_2 encoding T6SS genes enriched in forager versus nurse samples. The genomic island is only present in strain A7 of Gilli_2 in our genome database and is integrated in a conserved genomic region. Genes in green indicate T6SS genes, genes in pink depict strain-specific genes beloning to the same genomic island, and genes in grey are other other genes present in the flanking regions of genomes of Gilli_2. Genes enriched in forager samples are labeled in bold and flagged with an asterisk. **Fig. S10** (A) Actin gene copies in 10ng of each sample. (B) 16S rRNA gene copy number per actin gene copy per sample. (C) rRNA gene copy number per actin gene copy per ng of total DNA. The normalization with actin gene copies amplifies the differences between the two groups, as there are fewer copies in foragers than in nurses. **Fig. S11.** Cumulative length of all single-copy core genes (Core genome length, in green) relative to the total genome length of the reference strain (Total genome length, in blue) of each species. Reference strain is the strain used in the reduced reference database. The filtered core genome length represents the portion of the core genes which were used for the SNV analysis as they had >10x read coverage across all samples.

**Additional file 2: Table S1.** Sample collection information. **Table S2.** Table of all strain genomes included in the reference database, their Species/SDP abbreviation used in this study, their corresponding full species name (if existing) and the host where they were isolated. Genomes used as representative in the reduced database are marked by 'y' in the last column. **Table S3.** Proportions of reads mapping to species isolated in non-A. mellifera hosts. The proportions are computed as the number of reads mapping to the species, divided by the number of reads mapping to the same phylotype for this sample. **Table S4.** Table of differentially abundant orthogroups showing which behavioral state they were enriched in along with the associated fold-change, p-values, their COG category and functional

annotation. **Table S5.** Matrix of the counts of the OGs annotated as H category significantly enriched in the forager samples compared to the al OGs and all COG categories.

**Additional file 3.** Review history.

### Acknowledgements

We would like to thank Mam Malick Sy Ndiaye, Florent Mazel, Garance Sarton-Lohéac, and Shinichi Sunagawa for providing helpful comments on the manuscript. We would like to thank Julien Marquis and the team from the Lausanne Genomics Technology Facility for carrying out the Illumina sequencing. We would like to thank Benjamin Dainat, Jan Dudda, Olivier Emery, and Chloé Peillon for letting us sample bees from their apiaries.

### Review history

The review history is available as Additional file 3.

### Peer review information

### Authors' contributions

P.E. and G.B. were involved in conceptualization; G.B. and A.P. carried out the formal analysis, investigation, and data curation. P.E. was responsible for funding acquisition, supervision, and providing resources. The methodology was developed by K.E., G.B. and A.P. Visualization and validation were carried out by G.B., P.E., A.P., and K.E. The original manuscript was written by G.B. and P.E.; then, P.E., A.P., and K.E. contributed to reviewing and editing. All authors approved the final manuscript.

### Funding

holarship awarded to A.P., an ERC Starting Grant (MicroBeeOme) and a Swiss National Science Foundation SPIRIT grant (grant no. IZSTZ0_189496) both awarded to P.E., and the NCCR Microbiomes, a National Centre of Competence in Research, funded by the Swiss National Science Foundation (grant no. 180575).

### Availability of data and materials

The sequencing data is available under the NCBI BioProject ID PRJNA904667 [77]. The honeybee gut bacterial strain genome database is available for download on Zenodo: https://zenodo.org/records/10182034 [78]. The scripts required to run the analyses and produce the figures are available on GitHub: https://github.com/Aiswarya-prasad/Publication_Baud_metaG_NvsF_2023 [76] and Zenodo: https://zenodo.org/records/10183178 [79]. Both the repositories are released under the MIT license.

## Declarations

### Ethics approval and consent to participate

Not applicable.

### Competing interests

The authors declare that they have no competing interests.

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

## 
