## [**Additional file 3.** Review history. · Genome Biology]

Review History

First round of review

Reviewer 1

Are you able to assess all statistics in the manuscript, including the appropriateness of statistical tests used? Yes, and I have assessed the statistics in my report.

Comments to author:

The manuscript "Turnover of strain-level diversity modulates functional traits in the honeybee gut microbiome between nurses and foragers" by Baud et. al. investigates the differences in the gut microbiome between two behavioural states of honeybees: foragers and nurses. Both bee states are characterized by differences in behaviour, task and diet. The study investigates 15 beehives from 5 locations (3 beehives at each location) and pooled the guts of 20 bees for each beehive.

Overall, the authors discovered differences in species abundances profiles, total bacterial load, strain-diversity and -composition, and strain-specific gene content between foragers and nurses. Moreover, the differences in strain-specific gene content suggests that distinct strains represent 'ecologically relevant units' associated with their hosts' behavioural state.

The study is a valuable and relevant addition to the field of microbiome research as it shows that species-level and strain-level variation is both associated with host lifestyle, but both levels of resolution reveal different dynamics about the underlying processes. We can nicely see that to describe a host-associated microbiome it is not sufficient to profile the species, as fundamental differences in the functional capabilities are often found at the strain level.

Generally, the dataset and approaches in the paper are appropriate to address the questions discussed. However, there are some aspects that need clarification or adjustments in my opinion, mostly regarding the strain-level analysis (see points listed below).

Results

Line 116: "absolute abundance of bacteria relative to the amount of tissue" - as I understand the methods this is not entirely correct as the abundance was not measured to the tissue weight but towards host marker gene using qPCR. I would clarify to say "absolute abundance of bacteria relative to host DNA".

Line 152 + Fig. 1C, 1D: It would be good to test the difference between communities using a permanova.

Line 162: It would be helpful to support the statement that this phylotype is more even in foragers with an estimate of evenness between the two bee states, showing that it is indeed different.

Line 153-169: The authors acknowledge that the lower absolute abundance of species in the foragers is likely a consequence of the lower total bacterial load. While it is convincing to see that different species are affected to a different degree it might be interesting to see also the

change in relative abundances. I don't think it is necessary for the manuscript, though and understand if the authors prefer not to include this additional analysis.

Line 181-186: This part needs some clarification which was not entirely explained by the methods. What exactly is considered a polymorphic site? Is it if there is any alternative allele towards the reference (even if it is 100% of the reads) or is it only polymorphic if there is more than 1 allele in the reads. This is an important distinction because if also 100% alternative alleles towards the reference are considered polymorphic, then it makes a difference whether the reference genome was more closely related to the strain found in either foragers or nurses. For example, if there are 2 sites with 100% alternative allele towards the reference in the foragers, and these same two sites in the nurses correspond to the allele found in the reference it looks as if the foragers have a higher strain diversity. But they simply carry a different strain from the nurses while in both cases the strain diversity is equal. To measure the strain diversity in foragers vs. nurses only polymorphic sites with > 1 allele in the reads should be considered.

Line 206-209: This part needs some clarification and was not completely clear from the methods. To compare the polymorphic sites in sample pairs for each species, has the encoded dominant allele on the reads in the shared polymorphic sites been taken into account? For example, if two samples have the same 5 polymorphic sites but different dominant alleles, are they considered similar or dissimilar? I would think that if the question is 'how similar are the strain identities between 2 samples' then the allele is important to consider. I would address this question by looking at all detected polymorphic sites in a species and count the number of identical dominant alleles over the number of polymorphic sites as a measure of how similar the dominant strains are to each other. This way it can be tested whether the nucleotide identities within and between host behavioural types are different.

In the current analysis it looks to me as if only the positions of polymorphic sites were compared - and here it is unclear to me whether this included 100% alternative alleles towards the reference as mentioned in my point above. I think the current analysis done by the authors still gives a good estimate that strains in behavioural types are more similar to each other than between types, but to fully understand the information some clarifications on the methods would be helpful.

Line 209-217: I think it would be nice to include a perMANOVA model where the terms are added sequentially (distance \sim location + site + behavioural state). This way we could also see how much distance can be explained by behavioural state after accounting first for the other two variables (and the significance).

Line 237: From the methods I understand that to determine the ORF coverage (subsequently being summed to OG coverage) was determined by mapping the reads to the gene nucleotide sequences. If this is the case, it can be problematic as the read mapping towards the sequence edges usually drops therefore biasing shorter genes differently from longer genes. Like this a gene might look strain-specific simply because of these artefacts. It will be more reliable to extract the per-gene-region coverage from the reads mapping to the original contigs. If this is indeed what has been done, then the authors should specify it in the methods.

Line 293: only a single strain out of how many?

Discussion

Line 416-417: could one aspect be related to the size of surface area/epithelial area? Is this larger in forager guts and could this be related to a higher proportion of biofilms where these genes are needed?

Methods

Line 438: Is this the reason that for some beehives the numbers in Suppl. Table 1 are smaller than 20?)

Line 509-512: For which species this was the case?

Line 514-541: Clarification to the points raised above regarding the strain-level analysis

Data availability:

All the deposited data and resources are present as the authors describe

Figures

Fig. 1A: y-axis label should rather be 16S copy number per host marker copies?

Fig. 1E: It would be helpful to add somewhere in which beetype the abundance is higher for pos/neg log₂ fold change.

Fig. 2: Generally the grey background make it hard to see the points. It would be easier with a white background (or increase point sizes). Also, add to legend what permanova formula has been used. As per my comment above potentially consider a permanova with sequential addition of terms.

Supplementary Fig. 6: Specify permanova formula (what was tested)

Reviewer 2

Are you able to assess all statistics in the manuscript, including the appropriateness of statistical tests used? Yes, and I have assessed the statistics in my report.

Comments to author:

The authors present a strain-resolved metagenomic analysis of gut microbiomes of nurse and forager bees sampled from a geographically restricted area (Western Switzerland). They describe consistent strain-level differences between nurses and foragers within bacterial species, complementing previous work on species/phyloptype-level differences. They report consistent

trends both at the level of "similarity in shared polymorphic sites" (see below) and in individual genes or gene cassettes.

This is interesting and relevant work on an excellent model system to study such strain-level microbiome trends in hosts "in the wild". However, as detailed below, I have a series of comments and concerns: (i) several technical/methodological choices remain unclear and could be outlined/motivated more clearly; (ii) I feel that some of the claims in the title, introduction and discussion are not fully supported by the design and data; (iii) by design, the work remains descriptive, but there are several follow-up analyses that could significantly enhance the relevance of these findings beyond the studied model (bees).

Technical/Methodological comments:

* Ll. 208f and ll. 534f: the authors describe strain similarities based on a Jaccard index of "shared polymorphic sites" between (core) genomes. However, the motivation for this is unclear. I understand the Methods section such that the authors considered the observation of any variation relative to the reference at a given position (after applying cutoffs etc) between two samples. First, this definition of 'polymorphic' seems to ignore fully fixed SNPs relative to the reference (e.g., if the reference is A but 100% of reads in one sample at the same position are G) - such positions would still be highly relevant when comparing between samples. More importantly, this definition seems to ignore the actual allele frequencies at a site; in an extreme case, a 'polymorphic' site could be 80% A and 20% G in sample X, but 50% C and 50% T in a sample Y. As far as I understand, the authors would count such a signal as "shared polymorphic site" between samples X and Y which would clearly be incorrect. Why not simply compute the allele frequency similarity/dissimilarity across all considered sites between two samples? Apologies if I misinterpret what the authors did, but at least to me the Methods section in ll. 534f is not clear on this.

* Ll. 528f. Related to this: why were polymorphic sites with ≤ 0.1 alternate alleles removed? Given that 20 bees were pooled per sample and sequencing was quite deep, and given the authors' use of relatively conservative cutoffs, I would expect that a variant with a frequency of 10% (and depending on coverage even of 1%) would usually be robustly supported by the sequencing data and should not be discounted as noise. Rather than an overestimation of diversity I would argue that ignoring such sites/variants would mean an underestimation in fact.

* Ll. 514f. How did you account for reads mapping to multiple reference genes? How large was the fraction of such multiple mappers? Were they removed or otherwise accounted for? Given the outlined criteria (≥ 50 matches and ≤ 5 mismatches) it is possible that reads were included that mapped to reference sequences at up to 10% dissimilarity which may be non-specific between related species. Or is this the samtools view -F 0x800 command?

* Ll 494f. Why were core genes defined as "single copy" genes only? The usual definition of a "core" gene refers to its prevalence among different strains within a species/phylogroup (i.e., frequency in the pangenome), but it's unclear why an additional single copy criterion was used here.

* Ll. 514f. Please specify for each species/phylogroup how large the reference set of core genes was that was used for strain calling. How long was the concatenated length of all used genes in each case? What fraction of the ref genome did that correspond to?

- * L1 491f. "pfam from which BED files with gene positions were generated". This is unclear - pfam is not a "tool" and it can't be used to generate bed files with gene positions. Please clarify what was done here.
- * L1 494f. How was OrthoFinder parametrized? Does this refer only to 'novel' gene families that hadn't already been mapped by eggNOG as described in l. 491? Or were two different (independent?) systems of orthologs used here? If so, please clarify which set was used for which analyses.
- * ll. 128. 20.8% (and even 4.2%) of unmapped/unaccounted for reads in these relatively deeply sequenced metagenomes is still a substantial fraction. I suggest to taxonomically profile these for (other) prokaryotes using standard tools (MetaPhlan, mOTUs or even Kraken if need be), but also to check whether these reads are plant DNA (from pollen/nectar) which could provide additional relevant info.
- * L1. 215f: higher pseudo-F for behavior state over sampling site could be artefact (two states, but three sites); R2 would be more relevant here

Minor comments on interpretation/phrasing:

- * Is "turnover" in the title appropriate? There is no longitudinal data here to support that, even if the nurse and forager stages are sequential.
- * abstract: "the extent communities shift in strain composition in response to environmental changes has mostly remained elusive" - is that so? I think this discounts a substantial body of work, some of which the authors cite later on in the manuscript. Also: would the nurse to forager transition really be considered an "environmental change"?
- * "strains are ecologically relevant units" - this is not a novel statement or finding, even in metagenomics con and I suggest
- * L1 45f: not sure that (human) infant gut development is a good example in the context provided. Jumps from seasonal cycling (changing conditions) to ecological succession
- * l. 108: "eventual" should read "possible"

Further suggestions (at the authors' and editor's discretion...):

- * Given that the authors have additional information about the locations of the sampled colonies, I would be curious about geographical signals between the colonies at strain level. Stage (nurse vs forager) was a stronger signal than colony membership overall, but are there differences in the micro- and macrodiversity trends between these colonies, associated with geographic distance?
- * Are the differences in gene composition tied to specific genomic locations? Possibly even to genomic islands or plasmids that could be rapidly gained or lost? Fig 3C and Figure S8 anecdotally suggest this, but are there systematic trends (i.e., association with mobile elements)? Likewise, are SNVs clustered on the genome? Or more broadly, are we looking at ecology here (turnover of strains) or at (rapid, "reversible") evolutionary processes?

Rebuttal – Point-by-point reply

Reviewer reports:

Reviewer #1: The manuscript "Turnover of strain-level diversity modulates functional traits in the honeybee gut microbiome between nurses and foragers" by Baud et. al. investigates the differences in the gut microbiome between two behavioural states of honeybees: foragers and nurses. Both bee states are characterized by differences in behaviour, task and diet. The study investigates 15 beehives from 5 locations (3 beehives at each location) and pooled the guts of 20 bees for each beehive.

Overall, the authors discovered differences in species abundances profiles, total bacterial load, strain-diversity and -composition, and strain-specific gene content between foragers and nurses. Moreover, the differences in strain-specific gene content suggests that distinct strains represent 'ecologically relevant units' associated with their hosts' behavioural state.

The study is a valuable and relevant addition to the field of microbiome research as it shows that species-level and strain-level variation is both associated with host lifestyle, but both levels of resolution reveal different dynamics about the underlying processes. We can nicely see that to describe a host-associated microbiome it is not sufficient to profile the species, as fundamental differences in the functional capabilities are often found at the strain level.

Generally, the dataset and approaches in the paper are appropriate to address the questions discussed. However, there are some aspects that need clarification or adjustments in my opinion, mostly regarding the strain-level analysis (see points listed below).

REPLY: We thank the reviewers for taking the time to assess our manuscript and for the constructive feedback. In the following point-by-point reply, we explain how we have addressed each of the points. We believe the comments of the reviewer have helped to strengthen our manuscript. Thank you.

Results

Line 116: "absolute abundance of bacteria relative to the amount of tissue" - as I understand the methods this is not entirely correct as the abundance was not measured to the tissue weight but towards host marker gene using qPCR. I would clarify to say "absolute abundance of bacteria relative to host DNA".

REPLY: Thank you for raising this point. Although we measured actin gene copies, we did not use it for the final analysis, as we realized that there is a difference in the actin gene copy number per gut between the two groups (see Supplementary Figure 10A, below), amplifying the differences in bacterial loads between nurses and foragers (see Supplementary Figure 10B, below). Instead, we decided to normalize everything to the total DNA yield per sample and expressed the genome copies per 1ng of input DNA (see Suppl. Supplementary Figure 10C, same as in the original manuscript). This said, no matter how we normalized the data, nurses always had more bacteria in the gut than foragers, which is consistent with all previous studies (e.g. Kesnerova et al 2020). We have changed the legend of Figure 1 and edited the corresponding method section in the revised manuscript. For reasons of transparency and to justify our approach, we have also included the results of the alternative analysis (normalization with actin gene copies) in the method section of the revised manuscript on line 480-484 and as Supplementary Figure 10.

Supplementary Figure 10. **A**, Actin gene copies in 10ng of each sample. **B**, 16S rRNA gene copy number per actin gene copy per sample. **C**, 16S rRNA gene copy number per actin gene copy per ng of total DNA. The normalization with actin gene copies amplifies the differences between the two groups, as there are fewer copies in foragers than in nurses.

Line 152 + Fig. 1C, 1D: It would be good to test the difference between communities using a permanova.

REPLY: We tested the difference in community composition at the phylotype level (Fig. 1C) and species level (Fig. 1D) using the formula Distance ~ Location + Behavioural state. Both behavioral state and location were significant in explaining the difference in the community compositions at the phylotype and species levels. We included Location based on a comment from reviewer#2. However, we need to keep in mind that our study was not designed to test Location. It is confounded by Hive, as we have two samples per hive from each location.

We have included this information in the revised manuscript, on line 157-158 in the results section and on line 622-625 in the Methods section.

Line 162: It would be helpful to support the statement that this phylotype is more even in foragers with an estimate of evenness between the two bee states, showing that it is indeed different.

REPLY: Good point. The shift of the Firm5 species towards a less even composition in foragers has now been confirmed by comparing the Shannon index measured based on the proportions of these species (Wilcoxon signed-rank test, p-value = 8.8e-06).

We have included this information in the revised Supplementary Figure 3B for all multi-species phylotype and on line 622 in the Methods section.

Line 153-169: The authors acknowledge that the lower absolute abundance of species in the foragers is likely a consequence of the lower total bacterial load. While it is convincing to see that different species are affected to a different degree it might be interesting to see also the change in relative abundances. I don't think it is necessary for the manuscript, though and understand if the authors prefer not to include this additional analysis.

REPLY: The relative abundance data is shown in Figure 1B (stacked bar plot of all phylotypes/genera) and in Figure 1C and 1D (NMDS plots). However, to show the relative abundance of the species composition we have included separate stacked bar plots for each of the following phylotypes, Lactobacillus, Bombilactobacillus, Bifidobacterium, and Gilliamella, together with the alpha-diversity/evenness analysis. These plots are included in the revised Supplementary Figure 3.

Line 181-186: This part needs some clarification which was not entirely explained by the methods. What exactly is considered a polymorphic site? Is it if there is any alternative allele towards the reference (even if it is 100% of the reads) or is it only polymorphic if there is more than 1 allele in the reads. This is an important distinction because if also 100% alternative alleles towards the reference are considered polymorphic, then it makes a difference whether the reference genome was more closely related to the strain found in either foragers or nurses. For example, if there are 2 sites with 100% alternative allele towards the reference in the foragers, and these same two sites in the nurses correspond to the allele found in the reference it looks as if the foragers have a higher strain diversity. But they simply carry a different strain from the nurses while in both cases the strain diversity is equal. To measure the strain diversity in foragers vs. nurses only polymorphic sites with > 1 allele in the reads should be considered.

REPLY: We are sorry for not having provided this information in the methods section. A site is considered polymorphic in a given sample if there are at least two alleles at a frequency above the detection cutoff of 0.01 regardless of the identity of the reference allele. In consequence, the similarity of the reference to the sequences/strains found in nurses vs foragers did not influence our diversity estimates. We have included a corresponding sentence on Line 556-561, which reads as follows: "A position was considered polymorphic if more than one allele was detected within the metagenomic reads of that sample mapped to the reference regardless of what allele was in the reference genome." We also added to the results section (Line 560) that we only looked for polymorphisms within the mapped reads.

Line 206-209: This part needs some clarification and was not completely clear from the methods. To compare the polymorphic sites in sample pairs for each species, has the encoded dominant allele on the reads in the shared polymorphic sites been taken into account? For example, if two sample have the same 5 polymorphic sites but different dominant alleles, are they considered similar or dissimilar? I would think that if the question is 'how similar are the strain identities between 2 samples' then the allele is important to consider. I would address this question by looking at all detected polymorphic sites

in a species and count the number of identical dominant alleles over the number of polymorphic sites as a measure of how similar the dominant strains are to each other. This way it can be tested whether the nucleotide identities within and between host behavioural types are different. In the current analysis it looks to me as if only the positions of polymorphic sites were compared - and here it is unclear to me whether this included 100% alternative alleles towards the reference as mentioned in my point above. I think the current analysis done by the authors still gives a good estimate that strains in behavioural types are more similar to each other than between types, but to fully understand the information some clarifications on the methods would be helpful.

REPLY: We do not consider which alleles are dominant but only whether they are present at a frequency above the threshold of 0.01 and supported by more than 1 read. Positions where there is 1 allele at a frequency of 1 in both samples are not considered for comparison. Excluding such sites does miss cases where a different alternative allele is fixed in each sample. Including sites where there are fixed alleles lowers our resolution as it brings the distance estimates too close to each other. This is not surprising because, in the core region, most positions are fixed and shared. Hence, we only consider positions that are polymorphic in at least 1 sample in each pair for comparison. For example, if the alleles present in sample 1 are {A, T} and sample 2 {A, T} this site is counted as shared regardless of which the dominant allele is. However, if sample 2 has just {A}, this site is still considered for comparison as it is polymorphic for sample 1, but it is not counted as shared because sample 1 has a variant (T) that sample 2 does not have. In cases where there are different alternate alleles in each sample, i.e. sample 1 has {A, T} and sample 2 {A, G}, the site is included in the comparison but not considered shared. This is implemented in the script by considering those positions as shared that are polymorphic in at least one of the samples and have >1 allele in the intersection of their set of alleles detected we have added a more detailed explanation of our approach to the methods section 561-578.

Line 209-217: I think it would be nice to include a permanova model where the terms are added sequentially (distance ~ location + site + behavioural state). This way we could also see how much distance can be explained by behavioural state after accounting first for the other two variables (and the significance).

REPLY: We included a permanova model. However, our study was not designed to look at location (we sampled only three colonies per location and had two samples per colony per location), but rather at the effect of behavioral group. We have updated the results text about this point accordingly on line 218-221.

Line 237: From the methods I understand that to determine the ORF coverage (subsequently being summed to OG coverage) was determined by mapping the reads to the gene nucleotide sequences. If this is the case, it can be problematic as the read mapping towards the sequence edges usually drops therefore biasing shorter genes differently from longer genes. Like this a gene might look strain-specific simply because of these artefacts. It will be more reliable to extract the per-gene-region coverage from the reads mapping to the original contigs. If this is indeed what has been done, then the authors should specify it in the methods.

REPLY: We indeed have mapped reads to the gene catalogue and not the contigs. It is true that read mapping drops towards the sequence edges, which can be problematic for very short genes. This is exactly the reason why we have excluded ORFs of <300bp. Moreover, we have also considered reads that only mapped with 50bp of their length (150bp reads). As a consequence, we are certain that we have no bias in the gene catalogue mapping analysis. We have verified this by looking at the mean of per base coverage of the shortest genes in our analysis. We could not find any positive correlation between gene length and per base coverage for any of our samples (low Pearson's R values). Please find coverage vs gene length plots of two representative samples below.

Rebuttal Figure 1. The shortest genes have a similar per base read coverage as longer genes

Line 293: only a single strain out of how many?

REPLY: Good point. We have changed that sentence to: "...which was present in only one (strain A7) out of 17 strains of *Gilli_2* in our reference database"

Discussion

Line 416-417: could one aspect be related to the size of surface area/epithelial area? Is this larger in forager guts and could this be related to a higher proportion of biofilms where these genes are needed?

REPLY: This is indeed an interesting aspect. But we currently know little about it. So, we have no basis for such a discussion and would like to not include this point as we feel it is too speculative.

Methods

Line 438: Is this the reason that for some beehives the numbers in Suppl. Table 1 are smaller than 20?)

REPLY: Yes, this is the reason. Bees were sampled in the field and brought back to the lab. We did not want to repeat the sampling for the few colonies where we did not get 20 bees that showed signs of nurses vs foragers.

Line 509-512: For which species this was the case?

REPLY: This was the case for *bifido_1.1*, *bifido_1.2*, and *bifido_1.4*. We have added that info to the revised method section on line 531-533. For nearly all samples the other species in high prevalence showed a clear fit. From the plots summarizing coverage per base across representative genomes of these species, it is apparent that the lack of a clear fit is likely due to a lack of accurate synteny information or low coverage, which was seen in particular samples (for species that were low in prevalence as such) rather than incorrect mapping.

Line 514-541: Clarification to the points raised above regarding the strain-level analysis

REPLY: We have updated the methods accordingly. See above.

Data availability:

All the deposited data and resources are present as the authors describe

Figures

Fig. 1A: y-axis label should rather be 16S copy number per host marker copies?

REPLY: We changed that according to our reply to the first point of this reviewer.

Fig. 1E: It would be helpful to add somewhere in which beetype the abundance is higher for pos/neg log2 fold change.

REPLY: In the legend, we say "Log2 fold change in total normalized absolute abundance of bacterial species in forager samples relative to nurse samples from the same hive" and the y-axis it says (Foragers vs Nurses). In our opinion, this clearly indicates that it is abundance(F)/abundance(N).

Fig. 2: Generally the grey background make it hard to see the points. It would be easier with a white background (or increase point sizes). Also, add to legend what permanova formula has been used. As

per my comment above potentially consider a permanova with sequential addition of terms.
Supplementary Fig. 6: Specify permanova formula (what was tested)

REPLY: We have lightened the grey background in all panels and increased the point size in panel A.

Reviewer #2: The authors present a strain-resolved metagenomic analysis of gut microbiomes of nurse and forager bees sampled from a geographically restricted area (Western Switzerland). They describe consistent strain-level differences between nurses and foragers within bacterial species, complementing previous work on species/phylolevel differences. They report consistent trends both at the level of "similarity in shared polymorphic sites" (see below) and in individual genes or gene cassettes. This is interesting and relevant work on an excellent model system to study such strain-level microbiome trends in hosts "in the wild". However, as detailed below, I have a series of comments and concerns: (i) several technical/methodological choices remain unclear and could be outlined/motivated more clearly; (ii) I feel that some of the claims in the title, introduction and discussion are not fully supported by the design and data; (iii) by design, the work remains descriptive, but there are several follow-up analyses that could significantly enhance the relevance of these findings beyond the studied model (bees).

Technical/Methodological comments:

* LI. 208f and II. 534f: the authors describe strain similarities based on a Jaccard index of "shared polymorphic sites" between (core) genomes. However, the motivation for this is unclear. I understand the Methods section such that the authors considered the observation of any variation relative to the reference at a given position (after applying cutoffs etc) between two samples. First, this definition of 'polymorphic' seems to ignore fully fixed SNPs relative to the reference (e.g., if the reference is A but 100% of reads in one sample at the same position are G) - such positions would still be highly relevant when comparing between samples. More importantly, this definition seems to ignore the actual allele frequencies at a site; in an extreme case, a 'polymorphic' site could be 80% A and 20% G in sample X, but 50% C and 50% T in a sample Y. As far as I understand, the authors would count such a signal as "shared polymorphic site" between samples X and Y which would clearly be incorrect. Why not simply compute the allele frequency similarity/dissimilarity across all considered sites between two samples? Apologies if I misinterpret what the authors did, but at least to me the Methods section in II. 534f is not clear on this.

REPLY: Positions with one allele at a frequency of 1 in both samples are not considered for comparison. Excluding such sites does miss cases where a different alternative allele is fixed in each sample but this does not change the conclusion of our distance estimates. This is not surprising because, in the core region, most fixed positions are also shared in all sample pairs often there are none that are fixed and unique. Hence, we only consider positions that are polymorphic in at least 1 sample in each pair for comparison. For example, If the alleles present in sample 1 are {A, T} and sample 2 {A, T} this site is counted as shared regardless of which the dominant allele is. However, if sample 2 has just {A}, this site is still considered for comparison as it is polymorphic for sample 1, but it is not counted as shared because sample 1 has a variant (T) that sample 2 does not have. In cases where there are different alternate alleles in each sample, i.e. sample 1 has {A, T} and sample 2 {A, G}, the site is included in the comparison but not considered shared. This is implemented in the script by considering those positions as shared that are polymorphic in at least one of the samples and have >1 allele in the intersection of their set of alleles detected. We have included a more detailed explanation of our approach in the revised methods section line 561-578.

* LI. 528f. Related to this: why were polymorphic sites with ≤ 0.1 alternate alleles removed? Given that 20 bees were pooled per sample and sequencing was quite deep, and given the authors' use of relatively conservative cutoffs, I would expect that a variant with a frequency of 10% (and depending on coverage even of 1%) would usually be robustly supported by the sequencing data and should not be discounted as noise. Rather than an overestimation of diversity I would argue that ignoring such sites/variants would mean an underestimation in fact.

REPLY: We agree with the reviewer that our cutoff was rather conservative. Our aim with this cutoff was to avoid including too many rare variants. As foragers have less biomass in the gut, we could have more easily picked up rare variants in foragers than in nurses due to 'oversequencing'. However, we have carefully re-analyzed our data during the revision and found that based on rarefaction curves, lowering the detection cutoffs does not lead to the identification of more variants in foragers. We thus decided to

lower the frequency cutoff to 1% as suggested by the reviewer, but to include only alleles that are supported by at least 2 reads (to avoid including sequencing errors).

As a consequence, the % of polymorphic sites detected in a given sample increased for most species. As before, for some species, foragers had higher strain-level diversity than nurses. However, for other species, we observed the opposite trend. We have included these new observations in the revised results section (line 201) and discussion (line 361). Most importantly, as in the previous analysis, we find that the samples cluster by host behavioral state for many species based on pairwise Jaccard distances of shared polymorphic alleles. So, the main finding that nurses and foragers have distinct strain-level profiles holds across different SNV calling cutoffs, indicating that this is a very robust pattern.

We thank the reviewer for this really valuable comment and believe that the new analysis is now much more robust, and more accurately reflects the diversity present in our samples.

* LI. 514f. How did you account for reads mapping to multiple reference genes? How large was the fraction of such multiple mappers? Were they removed or otherwise accounted for? Given the outlined criteria (≥ 50 matches and ≤ 5 mismatches) it is possible that reads were included that mapped to reference sequences at up to 10% dissimilarity which may be non-specific between related species. Or is this the samtools view -F 0x800 command?

REPLY: Yes, it was the 'samtools view -F 0x800 command'. We have included this info in the methods on line 541.

* LI 494f. Why were core genes defined as "single copy" genes only? The usual definition of a "core" gene refers to its prevalence among different strains within a species/phylotype (i.e., frequency in the pangenome), but it's unclear why an additional single copy criterion was used here.

REPLY: Yes, in theory such gene families should also be considered core. But gene families containing paralogs can be problematic for coverage analyses, especially in cases where one of the two copies is a distantly related paralog not present in all genomes. The read coverage of such divergent paralogous genes may be different, especially if one of the two copies is not present in every genome. This interferes with the regression analyses. Very few core gene families had two copies in some of the genomes. So, few gene families had to be excluded. We now refer to the core genes, as single-copy core genes to avoid any misunderstanding.

* LI. 514f. Please specify for each species/phylotype how large the reference set of core genes was that was used for strain calling. How long was the concatenated length of all used genes in each case? What fraction of the ref genome did that correspond to?

REPLY: The summed length of the single-copy core genes corresponds to about 23-77% of the length of the genome of a given SDP. We have added this info as Supplementary Figure 11.

* LI 491f. "pfam from which BED files with gene positions were generated". This is unclear - pfam is not a "tool" and it can't be used to generate bed files with gene positions. Please clarify what was done here.

REPLY: We are sorry for the confusion. We have rewritten this part of the method section and hope that this is now more clear (503-514).

* LI 494f. How was OrthoFinder parametrized? Does this refer only to 'novel' gene families that hadn't already been mapped by eggNOG as described in l. 491? Or were two different (independent?) systems of orthologs used here? If so, please clarify which set was used for which analyses.

REPLY: EggNOG was used to annotate genes, not to assign them to ortholog groups (OGs). To generate OGs, we only used OrthoFinder with standard parameters. We have rewritten this method section and hope that this is now more clear (506-514).

* II. 128. 20.8% (and even 4.2%) of unmapped/unaccounted for reads in these relatively deeply sequenced metagenomes is still a substantial fraction. I suggest to taxonomically profile these for (other) prokaryotes using standard tools (MetaPhlan, mOTUs or even Kraken if need be), but also to check whether these reads are plant DNA (from pollen/nectar) which could provide additional relevant info.

REPLY: Excellent point. We profiled the unmapped reads using mOTUs2 (version 3.0.3), which provides relative abundances of operational taxonomic units (mOTUs) based on universal marker genes. A large fraction of the unmapped reads could indeed be assigned to bacterial genera (between 54% and 97% of

the unmapped reads per sample). Particularly abundant genera among the unmapped reads included *Lactobacillus*, *Snodgrassella* and *Bartonella* indicating that our reference genome database is not exhaustive for these specific bee-associated genera. However, we also detected a lot of other bacterial species some of which have previously been associated with honey bees, but are considered as accessory or transient members as their occurrence varies a lot across individual bees. This was also the case here. Interestingly, they were more often detected in nurse samples than in forager samples. We have included these new results in Supplementary Figure 2 and in the corresponding results section on 131-136. This info will be helpful for future culturing and genome sequencing approaches.

* LI. 215f: higher pseudo-F for behav state over sampling site could be artefact (two states, but three sites); R2 would be more relevant here

REPLY: We have updated this comparison to consider instead the adjusted R2 to account for the degrees of freedom and number of observations. However, as already explained above, our study was not designed to look at location, but rather at the effect of behavioral group. We have updated the results text about this point accordingly on line 222-223.

Minor comments on interpretation/phrasing:

* Is "turnover" in the title appropriate? There is no longitudinal data here to support that, even if the nurse and forager stages are sequential.

REPLY: Yes, we believe this is appropriate. In ecology the term turnover can be used to talk about beta-diversity on cross sectional data. It does not have to be longitudinal data. N and F samples came from the same hive and they represent sequential stages. So, consequently there must be a turnover/change of the strain-level composition given our results.

* abstract: "the extent communities shift in strain composition in response to environmental changes has mostly remained elusive" - is that so? I think this discounts a substantial body work, some of which the authors cite later on in the manuscript. Also: would the nurse to forager transition really be considered an "environmental change"?

REPLY: Yes, it is an environmental change for the microbiota, as the host is the immediate environment of the bee gut microbiota. However, we have changed this sentence as follows to tone down and to make clear that the environment can be a host (but also the 'free environment'): "*However, to what extent communities undergo consistent shifts in strain composition in response to environmental/host changes is less well understood*"

* "strains are ecologically relevant units" - this is not a novel statement or finding, even in metagenomics con and I suggest

REPLY: We agree and changed the final statement as follows: "Our findings show that strain-level diversity in host-associated communities can undergo consistent changes in response to host behavioral maturation modulating the functional potential of the community."

* LI 45f: not sure that (human) infant gut development is a good example in the context provided. Jumps from seasonal cycling (changing conditions) to ecological succession

REPLY: Yes, it is a very different example than the seasonal cycling, but it is an example of a more or less predictable community change over time. We would thus like to keep it.

* I. 108: "eventual" should read "possible"

REPLY: Changed as suggested.

Further suggestions (at the authors' and editor's discretion...):

* Given that the authors have additional information about the locations of the sampled colonies, I would be curious about geographical signals between the colonies at strain level. Stage (nurse vs forager) was a stronger signal than colony membership overall, but are there differences in the micro- and macrodiversity trends between these colonies, associated with geographic distance?

REPLY: Both behavioral state and location were significant in explaining the difference in the community diversity. However, we need to keep in mind that our study was not designed to test for the effect Location. It is confounded by Hive, as we have two samples, one nurse and one forager, per hive from

each location. The effect size of Location or colony membership cannot be accounted for accurately in our design.

* Are the differences in gene composition tied to specific genomic locations? Possibly even to genomic islands or plasmids that could be rapidly gained or lost? Fig 3C and Figure S8 anecdotally suggest this, but are there systematic trends (i.e., association with mobile elements)?

|

REPLY: We believe it is generally understood that differences in gene content between strains of the same species are mostly due to horizontal gene transfer of genomic islands of different size. We don't think that such an analysis will add much novelty. Also, it is technically quite challenging, as many assemblies break at mobile elements as they typically present repeats.

Likewise, are SNVs clustered on the genome? Or more broadly, are we looking at ecology here (turnover of strains) or at (rapid, "reversible") evolutionary processes?

REPLY: The fraction of polymorphic sites that we have detected is unlikely explained by rapid/reversible evolutionary processes. Also, many of the alleles we have detected are in fact shared by at least some samples, which suggests that they do not present random in situ mutations, but mutations that have emerged in ancestors of the identified stains.

Second round of review

Reviewer 2

I commend the authors on their revision of the present manuscript. In my opinion, almost all issues that had been raised have been addressed. I only remain unconvinced of the authors' calculation of strain-level distances based on a Jaccard index of shared polymorphic sites, also following the clarification and further explanation provided by the authors. However, I also do not think that the authors' approach is inherently wrong or would overly skew the results and interpretation, so this should not impede publication of the manuscript.